# Laws Governing Free and Actual Drying Shrinkage of 50 mm Thick Mongolian Scotch Pine Timber

**Xiaodong Zhu, Jingyao Zhao, Wanhui Gao, Cheng Qian, Yunjia Duan, Shuaichao Niu and Yingchun Cai \***

Ministry of Education, Key Laboratory of Bio-Based Material Science and Technology, College of Material Science and Engineering, Northeast Forestry University, Harbin 150040, China; zxd196517@163.com (X.Z.); zjy_20180328@nefu.edu.cn (J.Z.); gaowanhui0720@163.com (W.G.); qc870723458@163.com (C.Q.); a15663741101@163.com (Y.D.); 13101607001@163.com (S.N.)

\* Correspondence: caiyingchunnefu@163.com

**Abstract:** The relationships between free shrinkage and actual shrinkage of different layers in Mongolian Scotch pine (Pinus sylvestris var. mongolica Litv.) were explored to provide basic data for the further study of drying shrinkage properties. The free shrinkage coefficients at different temperatures and the actual shrinkage strain of each layer were examined under conventional drying. The results showed high precision of free drying shrinkage of corresponding layers of thin small test strips in each layer of sawn timber. The free shrinkage increased linearly as moisture content declined. At the same temperature, the free shrinkage coefficient reached the largest values for the first layer (above 0.267%), while the smallest values were recorded for the ninth layer (below 0.249%). Except for the ninth layer, the free shrinkage coefficients in width directions of other representative layers decreased as temperature increased. At constant temperature, the difference in free shrinkage coefficient of test materials in the length direction of sawn timber was small for the first layer, but slightly larger and changed irregularly in the fifth and ninth layer direction. At the end of conventional drying, the plastic deformation of each layer in the early stage of drying showed a reducing trend or even reversal due to the effects of reverse stress and later damp heat. In sum, these findings look promising for future optimization of wood drying process.

**Keywords:** Mongolian Scotch pine; free shrinkage; effect of temperature; actual shrinkage

## 1. Introduction

The accurate determination of sawing allowance and improvement in drying quality are critical parameters for the efficient utilization of timber. The non-synchronous drying shrinkage caused by the uneven distribution of moisture content in sawn timber coupled with the anisotropy of drying shrinkage leads to drying stress and drying defects [1–5]. Thus, determining the variation law of drying strain during the drying process is important to understand the drying stress, optimize the drying process, improve the drying quality, and shorten the drying time. Among existing strains, elastic strain reflects the nature (tension or compression) and magnitude of drying stress at a particular time. On the other hand, strains-based viscoelastic creep and mechanical adsorption creep reflect plastic deformation under drying stress. The former could be recovered after stress elimination with time, while the latter is a permanent deformation that cannot be recovered under conventional conditions [6–9]. Both strains, especially the latter, affect internal cracks in the later stages of the drying processes, thereby affecting the quality of the resulting products at the end of the drying process [10–15]. The actual drying shrinkage strains include elastic strain, viscoelastic creep strain, and mechanical adsorption creep strain. The actual drying shrinkage strain obtained when drying the product to target moisture content could accurately determine the machining allowance [16–18]. Free drying shrinkage of wood relies upon negligible drying stress of small thin specimens during slow drying (desorption). Here, the change in size is caused by the loss of water in the wood cell wall.

Therefore, it is important to study free drying shrinkage characteristics to figure out the fiber saturation point under corresponding temperature and humidity conditions, as well as the influence of temperature on free drying shrinkage properties of wood [19–26].

So far, few laws governing free drying shrinkage and the creep strain of wood have been studied. Dengyun et al. investigated the laws guiding the free shrinkage and rheology of Masson pine to determine a drying stress model [27]. Jianfeng et al. studied the free shrinkage deformation of larch and hemlock at different temperatures [28]. Lin et al. explored the free drying shrinkage law of Eucalyptus urophylla under different thickness conditions [29]. Almeida and Perre examined the free shrinkage at the level of wood cells [30,31]. However, these studies are limited by many issues. The first relies on reference size when free drying shrinkage to target moisture content is required to measure the mechanical adsorption creep strain. This is often obtained when the strain test piece is treated for moisture absorption to eliminate plastic deformation and dried to the target moisture content. However, it is still unclear whether the adsorption process could completely restore the plastic deformation. The second has to do with the study of free drying shrinkage performance. Small specimens are usually used to measure certain moisture contents below the fiber saturation point. The drying shrinkages in chord, radial, and fiber directions are used under absolutely dry conditions to calculate the drying shrinkage coefficients in all three directions. Nevertheless, ensuring that the above drying shrinkage as a free drying shrinkage is challenging since standard specimens used would inevitably produce stress under rapid drying conditions. As a result, differences exist in texture direction and moisture content of each layer along the thickness and width of sawn timber. The free drying shrinkage size of each layer can not be analyzed by sawn timber tangential, radial drying shrinkage coefficient, or average moisture content to accurately calculate the mechanical adsorption creep.

To solve the above problems, 50 mm thick sawn timber of Pinus was used in this work as a study object. Test strips of each layer decomposed from adjacent test pieces at both ends of conventional drying test timber were dried slowly under different temperature and humidity conditions. The free drying shrinkage performances of the corresponding layer in width direction on the thickness of the test strip at each temperature were identified by the mean value of the free drying shrinkage performance parameters of the test strip in the same layer. Conventional drying experiments were carried out on test materials by measuring the actual size, drying shrinkage, and moisture content of each layer on the thickness during drying to target moisture content. The difference between the actual drying shrinkage and free drying shrinkage, as well as the influence of the difference on the drying stress, were all analyzed. Basic data for the accurate calculation of mechanical adsorption creep of each layer in different stages of sawn timber drying process of Pinus were provided, along with the theoretical basis for optimizing the wood-drying process. In sum, these findings provided a reference method for accurate determination of wood processing allowance.

## 2. Methods

### 2.1. Materials and Instruments

The small-diameter Pinus from the Greater Khingan Range forest area was processed into a size 4000 mm in length (fiber direction), 200 mm in width (near tangential direction of the upper surface layer and the near radial direction of the lower surface layer) and 50 mm in thickness (radial) to yield chord cut plate sawn timber with an initial moisture content of about 90%. The sawn timber was used to cut four 500 mm long conventional drying test materials used for the actual drying shrinkage performance, as well as free drying shrinkage performance test strips at both ends (six for each without obvious defects). The dimension of the test pieces along the length of sawn timber was fixed to 10 mm (fiber direction) $\times$ 200 mm $\times$ 50 mm. The test pieces were then decomposed into nine layers along the thickness of sawn timber to obtain test strips used for the free shrinkage performance of each layer in the thickness direction of the test timber. The layers were

numbered sequentially from the upper surface layer in the near-chord direction to the lower surface layer in the near-radial direction.

### 2.2. Calculation of Free Drying Shrinkage Performance

The free drying shrinkage of wood consisted of drying shrinkage without the action of internal and external stress. To ensure no-stress free drying shrinkage test strips during drying, the specimens were processed into thin strips of 10 mm (fiber direction) × 200 mm × 5.5 mm (Figure 1). A DHS-225 constant temperature and humidity drying kiln was employed to slow dry the test strips under the temperature and humidity conditions listed in Table 1. Next, the weight and length of each test strip were measured once dried to desorption stable moisture content under each temperature and humidity condition, meaning slightly higher than the corresponding equilibrium moisture contents listed in Table 1. The moisture content and free shrinkage of test strips were then calculated using Equations (1) and (2), respectively. After the last weight and size tests were carried out at the lowest equilibrium moisture content and same temperature, the samples were heated to absolute dry at 103 ± 2 °C, and the absolute dry weights were obtained. To study the effects of temperature on free drying shrinkage, dry bulb temperatures were set to 60, 80, and 100 °C, respectively. Note that the drying object under different humidity conditions at each temperature consisted of a test strip of each layer decomposed by two test pieces at both ends of each conventional drying test sample. This meant 90 test strips decomposed by 10 test pieces, including common test pieces at both ends of 4 test materials. The free drying shrinkage performance parameters of each layer in test materials represented the mean values of the parameters of 4 test strips in corresponding layers of 4 test pieces at both ends.

**Table 1.** Drying conditions of test strips used for free drying shrinkage performance tests.

| T (°C) | ΔT (°C) | RH (%) | EMC (%) |
|---|---|---|---|
| 60 | 0 | 100 | 27 |
| | 1 | 95 | 21 |
| | 3 | 86 | 15.5 |
| | 7 | 69 | 10.5 |
| | 13 | 49 | 7 |
| | 17 | 30 | 4.5 |
| 80 | 0 | 100 | 25 |
| | 1 | 96 | 19 |
| | 3 | 88 | 14.5 |
| | 7 | 73 | 10 |
| | 13 | 55 | 7 |
| | 17 | 39 | 5 |
| 100 | 0 | 100 | 22 |
| | 1 | 97 | 16.5 |
| | 3 | 90 | 13 |
| | 7 | 77 | 9.5 |
| | 13 | 61 | 6.5 |
| | 17 | 45 | 4.5 |

Note: The symbols T, ΔT, RH and EMC represent the dry bulb temperature, temperature difference between dry and wet bulb, relative humidity and equilibrium moisture content, respectively.

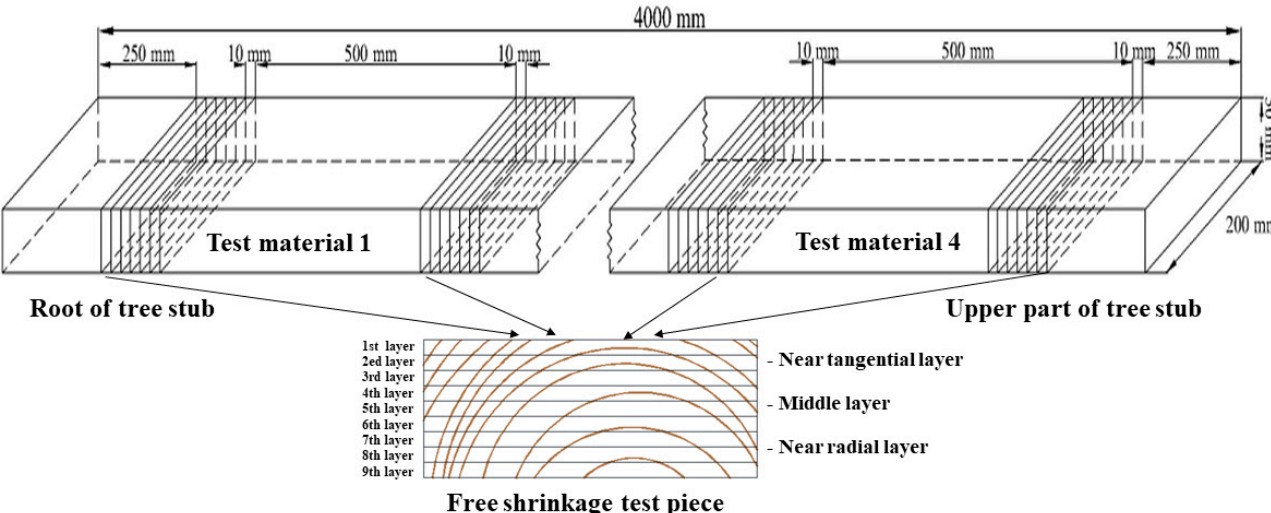

**Figure 1.** Preparation of specimens used to study the free and actual drying shrinkage laws of Mongolian Scotch pine sawn timber.

The water contents of Mongolian Scotch pine test material and strip were estimated by Equation (1):

$$MC_t = \frac{G_t - G_d}{G_d} \times 100\% \tag{1}$$

where $MC_t$ is the moisture content of test strip at time t; $G_t$ refers to weight of test strip at time $t$, and $G_d$ is absolute dry weight of test strip.

The free drying shrinkage of each test piece was calculated according to Equation (2):

$$y_{mc} = \frac{L_0 - L_{mc}}{L_0} \times 100\% \tag{2}$$

where $y_{mc}$ is the drying shrinkage at moisture content of the test strip $MC$; $L_0$ represents the initial length, and $L_{mc}$ is length at water content $MC$.

The free shrinkage coefficient would be the ratio of change in free shrinkage under fiber saturation point to the corresponding change in moisture content. This can be calculated using Equation (3):

$$K = \frac{y_{mc}}{FSP - 100 \times MC} \tag{3}$$

where $K$ is the free shrinkage coefficient; $MC$ refers to moisture content at time t below the saturation point of test strip fiber; $y_{mc}$ denotes the free shrinkage at the corresponding moisture content $MC$, and $FSP$ is the fiber saturation point of the corresponding layer test strip.

The drying shrinkage coefficient would represent the slope of the relationship curve between free drying shrinkage and water content obtained from experiments. This can be determined by the slope of the fitting of the curve.

### 2.3. Actual Drying Shrinkage Tests of Mongolian Scotch Pine Sawn Timber

The actual drying shrinkage tests of Mongolian Scotch pine sawn timber were carried out using four conventional drying test materials (Figure 1). A DHS-225 constant temperature and humidity drying oven was utilized for common conventional drying standards shown in Table 2. At average moisture contents of about 28% and 24%, two 10 mm shrinkage test pieces were cut from 100 mm away from the end face. Afterward, the two parts other than the test piece were bonded with silica gel and further dried until the target moisture content declined to 10%. Next, two 10mm test pieces were cut from the middle of each test piece along its length direction, and two 100 mm ends were removed. For cutting the test piece, the layer line along the thickness direction of the sawn wood

was immediately divided. The test strips of each layer corresponded to the layered test strips of a test piece used for free shrinkage performance tests at both ends of the test pieces cut before drying. The length of each layer decomposed along the scribed line was then measured, and the moisture content of each test strip at that time was evaluated by drying and weighing. Next, the corresponding length of the layered test strip before drying was tested according to the length of each layer in the test strip before decomposition and free shrinkage performance adjacent to both ends of the test material. The actual drying shrinkage rate of each layer in the test material was calculated as a function of time. The mean value of the corresponding layer of the three parts was estimated according to Equation (2). The free drying shrinkage of each layer to the free shrinkage coefficient, as well as the current water content of each layer, was calculated according to Equation (3). The free shrinkage size of each layer of the test strip at that time was calculated according to Equation (4).

$$L_{mcf} = \frac{100 - y_{mcf}}{100} \times L_0 \tag{4}$$

where $L_{mcf}$ represents the theoretically calculated length of free drying shrinkage of each layer in test strip during actual drying shrinkage performance testing at moisture content $MC$; $L_0$ is the initial length, and $y_{mcf}$ denotes the free drying shrinkage at moisture content $MC$.

**Table 2.** Schedule used for conventional drying of Mongolian Scotch Pine.

| MC/% | T/°C | ΔT/°C | RH/% | EMC/% |
|---|---|---|---|---|
| above 40 | 65 | 3 | 86 | 15.0 |
| 40~30 | 67 | 4 | 82 | 13.5 |
| 30~25 | 70 | 6 | 76 | 11.1 |
| 25~20 | 75 | 8 | 70 | 9.5 |
| 20~15 | 80 | 14 | 53 | 6.5 |
| below 15 | 90 | 25 | 32 | 3.8 |

## 3. Results and Discussion

### 3.1. Free Drying Shrinkage Properties of Different Layers in Mongolian Scotch Pine Chord Cut Plate

The relationship between free drying shrinkage and moisture content in the direction of strip length (specimen width) of adjacent specimen representative layers (1st, 5th, and 9th) at both ends of Mongolian Scotch pine test materials at 60, 80, and 100 °C are provided in Figure 2. The free shrinkage increased linearly as water content decreased. The slope of the relationship curve (fitting line) corresponded to shrinkage coefficient K; the ordinate intercept represented the total drying free shrinkage (maximum free shrinkage) $y_{max}$, and the abscissa intercept referred to the fiber saturation point (FSP). The mean and deviation of the above three parameters at all three temperatures determined by the above fitting curve equation of the representative layer test strip of adjacent test pieces at both ends of test material 1 are listed in Table 3. The free drying shrinkage properties of each layer in the width direction of the thickness in Pinus were evaluated using the above three parameters of the corresponding layer of the test strip of adjacent test pieces at both ends.

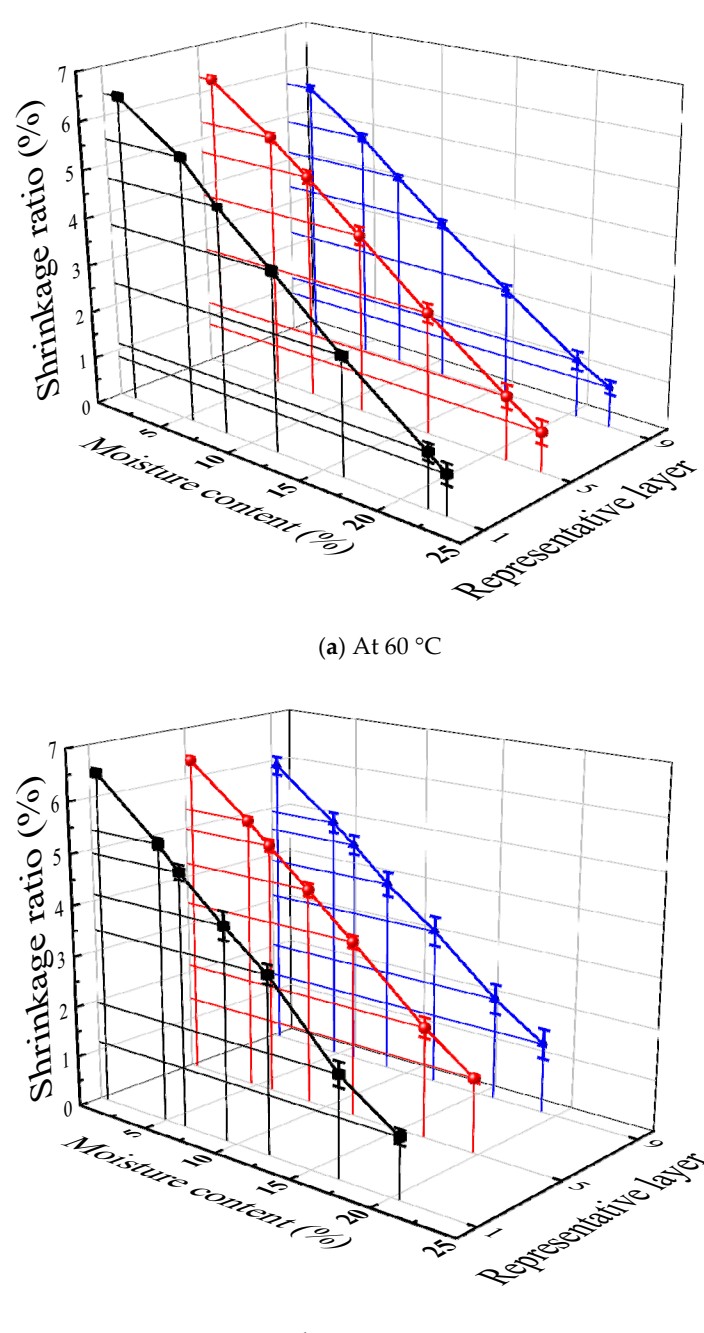

(**a**) At 60 °C

(**b**) At 60 °C

**Figure 2.** *Cont.*

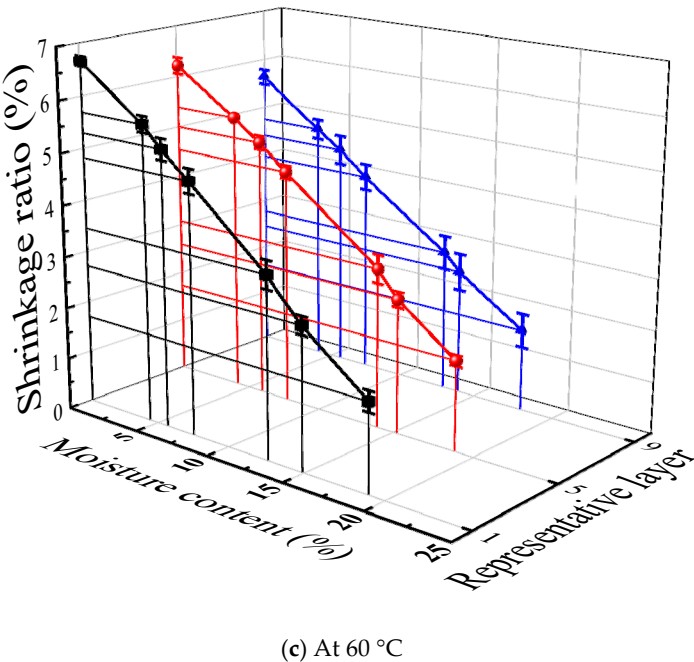

(**c**) At 60 °C

**Figure 2.** Relationship between free shrinkage and moisture content of representative layer strips in Mongolian Scotch pine at different temperatures.

**Table 3.** Fitting parameters of the free drying shrinkage law of representative layer test strips in Mongolian Scotch pine sawn timber.

| T (°C) | Representative Layer | K (%) | K Deviation | $y_{max}$ (%) | $y_{max}$ Deviation | FSP (%) | FSP Deviation | Coefficient of Determination ($R^2$) |
|---|---|---|---|---|---|---|---|---|
| 60 | 1st layer | 0.282 | 0.0048 | 6.881 | 0.0442 | 24.357 | 0.4271 | |
| | 5th layer | 0.264 | 0.0045 | 6.733 | 0.5806 | 25.471 | 0.5063 | ≥0.9868 |
| | 9th layer | 0.248 | 0.0069 | 6.134 | 0.9517 | 24.814 | 0.6999 | |
| 80 | 1st layer | 0.275 | 0.0082 | 6.693 | 0.1570 | 24.343 | 0.5473 | |
| | 5th layer | 0.261 | 0.0035 | 6.541 | 0.1141 | 25.086 | 0.2850 | ≥0.99089 |
| | 9th layer | 0.249 | 0.0047 | 6.064 | 0.1538 | 24.314 | 0.8408 | |
| 100 | 1st layer | 0.267 | 0.0024 | 6.797 | 0.1000 | 25.400 | 0.2777 | |
| | 5th layer | 0.244 | 0.0058 | 6.420 | 0.1343 | 26.386 | 0.3314 | ≥0.98782 |
| | 9th layer | 0.227 | 0.0047 | 5.700 | 0.1346 | 25.143 | 0.7404 | |

Note that T represents the temperature. K, $y_{max}$, and FSP are, respectively, the free shrinkage coefficient, the total drying free shrinkage rate, and the fiber saturation point of the Pinus sylvestris specimen used in experiments.

In Table 3, the minimum value of the determined coefficient of the fitting equation was 0.9868, thereby approaching 1 and showing high correlation. This indicated the high accuracy and reliability of the fitting in describing the corresponding free shrinkage performance. The deviations in free shrinkage coefficient and total drying shrinkage rate obtained by fitting were small, suggesting lower errors in free shrinkage performance differences and measurements of corresponding layers on the thickness of adjacent test pieces at both ends. The mean value of both parameters was used to characterize the free shrinkage properties of the corresponding layers in test materials and the data showed high accuracy.

3.1.1. Effects of Dry Shrinkage Anisotropy and Temperature on Free Drying Shrinkage Properties in Different Layer Width Directions of Test Material Thickness

The drying shrinkage coefficient in the width direction of a representative layer on the thickness of the test material at different temperatures are summarized in Figure 3. The free shrinkage coefficient of the first layer showed the largest values but with a decreasing trend. The differences in drying shrinkage coefficients between the first and ninth layers at 60, 80,

and 100 °C were estimated to 0.035%, 0.026%, and 0.041%, respectively. The dry shrinkage anisotropy of different layers in Pinus reached the largest value at 100 °C, while little and irregular influences on dry shrinkage anisotropy were observed at the other temperatures.

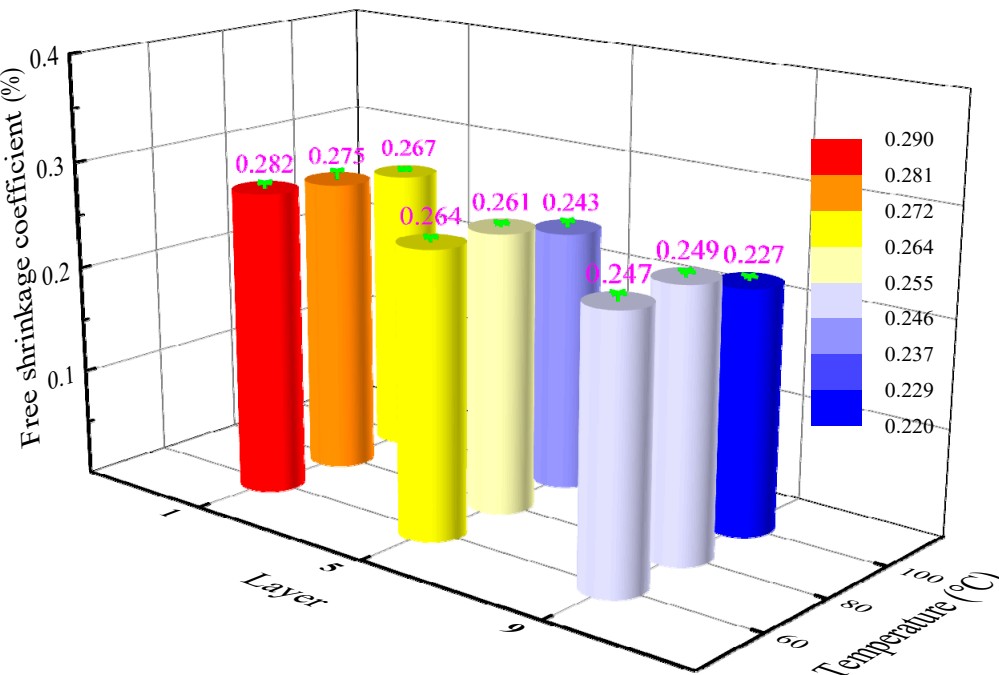

**Figure 3.** Effect of temperature on free shrinkage coefficient in the width direction of a representative layer on the thickness of the test material.

Except for the 9th layer, the free drying shrinkage coefficient in the width direction of other layers decreased as temperature rose, consistent with the law described by Jianfeng et al. [28] showing a decline in free drying shrinkage coefficient of larch with temperature increase. The 9th layer enhanced slightly after temperature rose from 60 to 80 °C followed by a decrease as temperature further rose. The differences in the free shrinkage coefficient of representative layers (1st, 5th, and 9th) on the thickness of the test material at 60 and 100 °C were determined to be 0.015%, 0.021%, and 0.022%, respectively. Hence, the effect of temperature on the free shrinkage coefficient of the near-tangential layer was lower than that of the near-radial layer [3]. The influence of temperature on the free drying shrinkage coefficient of Mongolian Scotch pine revealed that the temperature increase probably caused curl between cellulose chain molecules, as well as enhancement in arrangement spacing between cellulose molecular chains, thereby declining free dry shrinkage coefficient as temperature rose [7,9,30,31]. However, the exact reasons would require further studies.

3.1.2. Effects of Tree Stump Height on Free Drying Shrinkage Properties in Different Layer Width Directions of Test Material Thickness

The mean value and deviation of free shrinkage coefficient in the width direction of a representative layer on the thickness of the adjacent test piece at both ends of four sections of Pinus sawn timber in the length direction (Figure 1, tree stump) are gathered in Table 4. At the same temperature, the free shrinkage coefficient in the width direction of the representative layer on the thickness of different parts in the sawn timber of Pinus looked close to that obtained for the first layer. By comparison, the difference between the 5th and 9th layers looked slightly large and irregular with a maximum difference of 0.022% This may be due to the existence of more pith centers in the near-radial layer of sawn timber, as well as the poor straightness of the texture in the chord direction.

**Table 4.** Free drying shrinkage coefficient of a representative layer on the thickness of the test material at different parts in the length direction of Mongolian Scotch pine sawn timber.

| T (°C) | Layer | *k* of TM-2 (%) | Deviation | *k* of TM-3 (%) | Deviation | *k* of TM-4 (%) | Deviation |
|---|---|---|---|---|---|---|---|
| | 1st | 0.269 | 0.0063 | 0.273 | 0.0019 | 0.263 | 0.0020 |
| 60 | 5th | 0.276 | 0.0024 | 0.257 | 0.0042 | 0.264 | 0.0011 |
| | 9th | 0.261 | 0.0087 | 0.239 | 0.0015 | 0.254 | 0.0037 |
| | 1st | 0.276 | 0.0057 | 0.261 | 0.0012 | 0.265 | 0.0050 |
| 80 | 5th | 0.279 | 0.0025 | 0.255 | 0.0046 | 0.259 | 0.0062 |
| | 9th | 0.252 | 0.0045 | 0.230 | 0.0076 | 0.251 | 0.0102 |
| | 1st | 0.267 | 0.0060 | 0.268 | 0.0020 | 0.263 | 0.0083 |
| 100 | 5th | 0.248 | 0.0056 | 0.245 | 0.0107 | 0.256 | 0.0045 |
| | 9th | 0.229 | 0.0077 | 0.226 | 0.0109 | 0.245 | 0.0042 |

Note: The symbols k of TM-2(TM-3, TM-4) represent the value of the free shrinkage coefficient of the test material 2, 3 and 4, respectively. Test material 2 is near the root, and test material 4 is near the crown.

### 3.2. Comparison of Free Drying Shrinkage and Actual Drying Shrinkage in the Width Direction of Each Layer on the Thickness of Pinus Test Material

The relationship between the actual drying shrinkage and the free drying shrinkage of different layers on the thickness of Pinus at various moisture contents (early-stage I with average moisture content of 28%, early-stage II with average moisture content of 24%, and the end of conventional drying with average moisture content of 10%) are presented in Figure 4. The free drying shrinkage rate was obtained from the relationship between the above free drying shrinkage coefficient and temperature (Equation (2)). The difference between possible size during the drying of the test material and the corresponding moisture content in the width direction of different layers at various stages (early-stage I, early-stage II and the end of the drying period) and the actual size of the test material dried at corresponding moisture contents (difference in size after free and actual drying shrinkage) is shown in Figure 5. Note that the trend visually described the changes (deformations) in size under drying stress. The ratio to the original size consisted of drying strain composed of elastic strain, viscoelastic creep strain, and mechanical adsorption creep strain generated under drying stress. The percentage represented the difference between the actual drying shrinkage rate and free drying shrinkage rate shown in Figure 4.

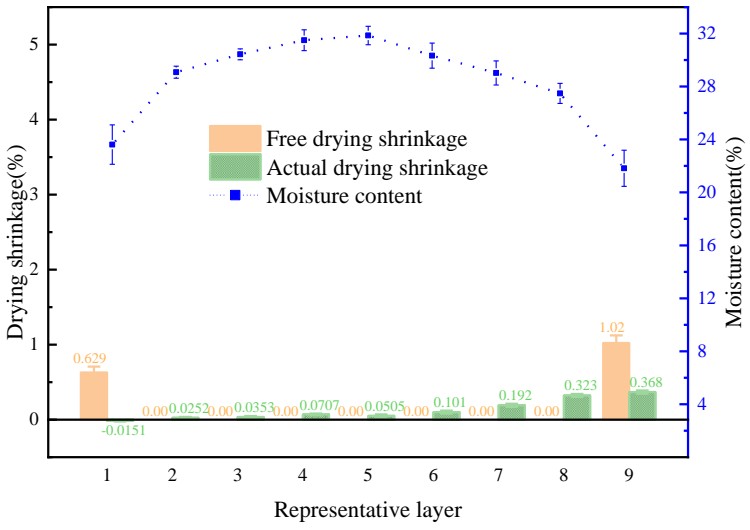

(**a**) Early drying stage I

**Figure 4.** *Cont.*

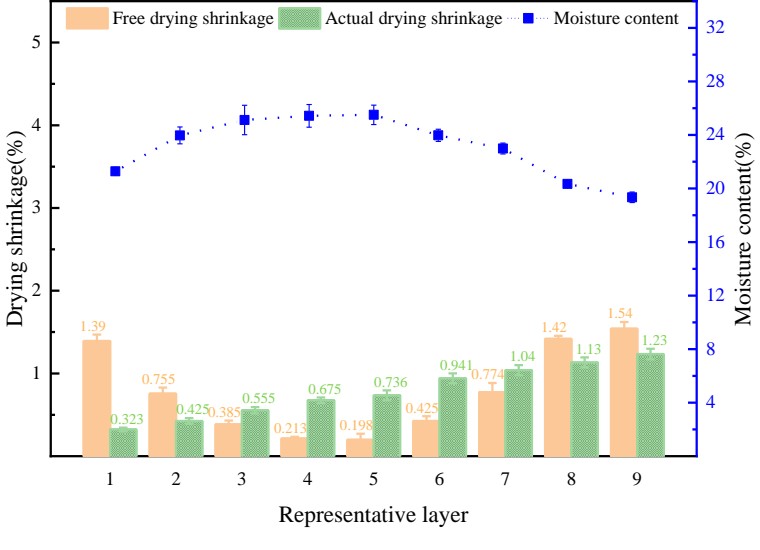

(**b**) Early drying stage II

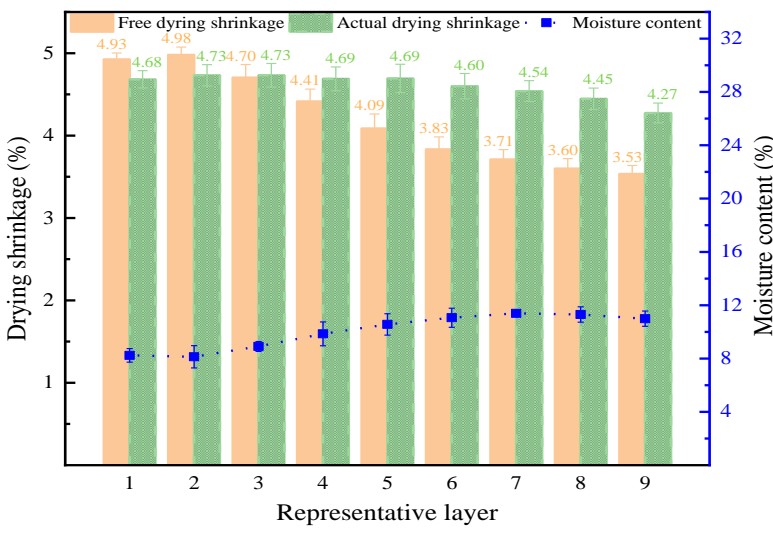

(**c**) End of drying

**Figure 4.** Comparison of the free and actual drying shrinkage of each layer in the thickness of Mongolian Scotch pine at each stage of conventional drying.

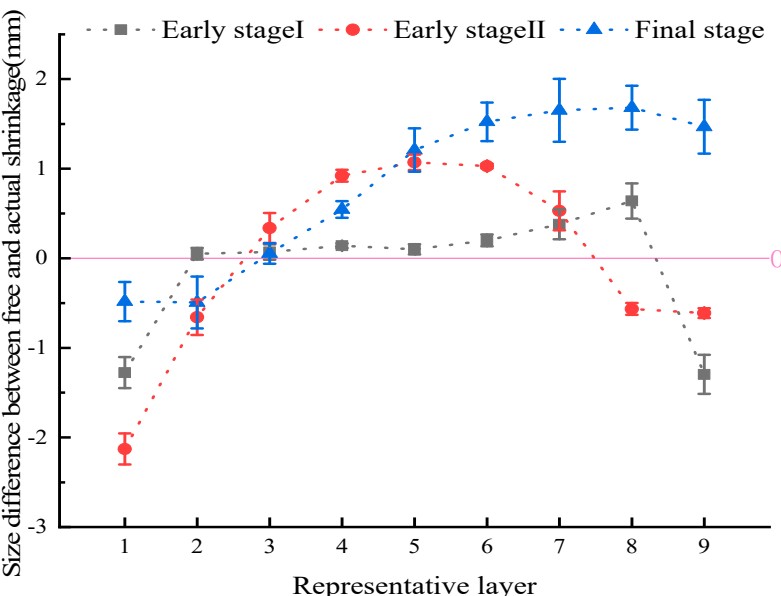

**Figure 5.** Size difference between free and actual dry shrinkage of each layer in the thickness of Pinus sylvestris test material. Note that the size difference between free shrinkage and actual shrinkage would equal the width when free shrinkage reached the corresponding moisture content and actual width when drying to the corresponding moisture content.

The moisture content of the surface layer (1st and 9th) was lower than FSP under the conventional drying process at early stage I (Figure 4). Note that the free drying shrinkage would reduce to free drying shrinkage under no influence by internal and external forces. Meanwhile, the free shrinkage rate of the 9th layer was greater than that of the 1st layer because of the lower water content in the near-radial layer when compared to the near-tangential layer, despite the moderate free shrinkage coefficient of the near-radial layer than the near-tangential layer. In fact, the moisture content of each internal layer higher than FSP led to no drying shrinkage with a free drying shrinkage rate of zero. This inhibited the drying shrinkage of the surface layer, resulting in tensile stress on the surface layer. In other words, the 9th layer with a high free drying shrinkage rate was greater than the 1st layer, leading to inside compressive stress. Moreover, the actual shrinkage rate of the 9th layer was the highest at that time but decreased slightly from the 9th layer to the 1st layer since the test material was not pressed to prevent bending during drying combined with bending under internal compressive stress (convex surface in 1st layer and concave surface in 9th layer) and unequal tensile stress of the upper and lower surface layers. At that time, the actual drying shrinkage rate of the surface layer (1st and 9th) of the test material was lower than the free drying shrinkage rate. Thus, the tensile deformation and deformation amount (negative value) were generated at this part under the action of tensile stress. By comparison, compressive strains and deformation amounts of other layers under compressive stress were positive (Figure 5).

During drying to early-stage II, the moisture content of each layer was lower than FSP (Figure 4b). If not affected by internal and external forces, each layer drying shrinkage ratio would shrink freely according to Figure 4. The moisture content decreased from the high-value central layer to the surface layer. Moreover, free shrinkage at the corresponding moisture content increased from the low-value central layer to the surface layer. In addition to the surface layer, the actual drying shrinkage rate of the subsurface layer (2nd and 8th) was also less than the free drying shrinkage rate. Besides, the actual dry shrinkage rate of each internal layer was greater than the free drying shrinkage rate. The changes in deformation from early-stage I drying to early-stage II are depicted in Figure 5. From early-stage I drying to early-stage II, the parts producing tensile stress and tensile deformation (negative value) extended from the surface to the subsurface layers, while the parts producing com-

pressive stress and compressive deformation (positive value) shrunk to inner layer. The increase in the tensile deformation of the 1st layer indicated the continuously restrained internally drying shrinkage of the layer, thereby bearing the tensile stress. By comparison, the decrease in the tensile deformation of the 9th layer indicated the inhibition of the drying shrinkage to the adjacent inner layers by the layer with tensile deformation in early-stage I. This produced compressive stress in the subsequent drying processes. The compression deformation of all four layers increased. Thus, the interior continued inhibiting the dry shrinkage of the surface layer while still bearing the compressive stress [2].

At the end of drying the process (Figure 4c), the moisture content in each layer of the test material was lower when compared to that in the near-radial direction. Moreover, the free drying shrinkage coefficient of each layer decreased from the near-tangential layer to the near-radial layer. Therefore, the free drying shrinkage was larger in the near-tangential direction and smaller in the near-radial direction. The actual shrinkage rates of layers in the test material showed little difference due to the overall restraining of each layer. Except for the 1st and 2nd layers (Figures 4c and 5), the free shrinkage rates at the end of drying near the tangential direction were greater than the actual shrinkage rates. Moreover, tensile deformation still existed, and the free shrinkage rates of other parts were less than the actual shrinkage rates with corresponding compression deformation. Hence, a series of research results related to the analysis of elastic strain and creep strain at different stages of the drying process of the same test material were carried out, and the results will be published separately. Figures 4a,b and 5 show changes in dry shrinkage difference and deformation from early-stage I drying to early-stage II drying. The free shrinkage coefficient of each layer decreased from the near-tangential layer to the near-radial layer. The variation in moisture content of each layer is depicted in Figure 4. The comprehensive analyses revealed that the tensile deformations in the early stage of the drying of the upper and lower surfaces, as well as the subsurface of the test material, were caused by tensile stress. The compression deformation under internal lamination stress contained a certain degree of plastic deformation (mechanical adsorption creep). At the end of drying, the shrinkage of the inner layer was restrained by the surface layer with tensile plastic deformation, resulting in tensile stress. Meanwhile, the surface layer showed compressive stress. Under the action of reverse stress and damp heat, the plastic deformations of the surface and internal layers in early-stage recovered to some extent. Even the tensile plastic deformations of the 8th and 9th layers illustrated compressive deformations under the action of damp heat and compressive stress at the later stage.

## 4. Conclusions

An accurate detection method of free shrinkage performance in different layer width directions of sawn timber thickness in Mongolian Scotch pine was successfully developed. The correlation coefficient determined from the regression equation of the curve was close to 1, conducive to the detection of free shrinkage performance in other directions.

The free dry shrinkage coefficient in the width direction of the representative layer on the thickness of the Pinus test material decreased from the near-tangential surface layer (1st layer, above 0.267%) to the near-radial surface layer (9th layer, below 0.249%) due to the influence of dry shrinkage anisotropy. The difference in the free dry shrinkage coefficient between the 1st layer and 9th layer reached a maximum at 100 °C (0.022%). Except the near-radial layer, with the increase of temperature, the free shrinkage coefficients of the first and ninth layers decreased from 0.282% and 0.264% to 0.267% and 0.243%, respectively. The experimental data at all three temperatures did not suffice to accurately determine the functional relationship between both. Thus, further future studies at various temperatures are required.

The above free shrinkage coefficients of different parts in the length direction of sawn timber, the middle layer and the near-radial layer showed irregular changes due to the influence of uneven texture. However, the near-tangential layer illustrated little difference.

At an average moisture content of around 28%, the surface of the test material was subjected to tensile stress, tensile elasticity and plastic deformation to yield a negative size difference between free and actual shrinkage. The internal layers were subjected to compressive stress, compressive elastic and plastic deformation to yield a positive size difference between free and actual dry shrinkage. As drying continued to reach an average moisture content of about 24%, the range of tensile stress, tensile elasticity and plastic deformation extended from the surface layer to the subsurface layer. By comparison, the range of compressive stress, compressive elasticity and plastic deformation showed shrinkage to the inner layer. At the end of drying, the plastic deformation of each layer in the early stage of drying reduced or even reversed due to reverse stress and damp heat effect in the subsequent drying process.

In sum, as thickness of sawn timber was strictly required during processing and utilization, studying the influence of sawn timber drying process conditions on the actual drying shrinkage in the thickness direction could provide great basis for future research directions.

**Author Contributions:** Formal analysis, X.Z.; investigation, C.Q.; resources, J.Z. and Y.C.; data curation, X.Z.; W.G.; C.Q.; Y.D. and S.N.; writing—original draft preparation, X.Z.; writing—review and editing, Y.C.; supervision, J.Z.; funding acquisition, Y.C. All authors have read and agreed to the published version of the manuscript."

**Funding:** This research was funded by the National Natural Science Foundation of China (Grants Nos. 31901242 and 31670562) and Heilongjiang Science Foundation Project (Grant No. LH2020C038).

**Institutional Review Board Statement:** Not applicable.

**Informed Consent Statement:** Not applicable.

**Data Availability Statement:** Not applicable.

**Acknowledgments:** This work was financially supported by the National Natural Science Foundation of China (Grants Nos. 31901242 and 31670562) and Heilongjiang Science Foundation Project (Grant No. LH2020C038).

**Conflicts of Interest:** The authors declare no conflict of interest.

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
