# Peer review of "Laws Governing Free and Actual Drying Shrinkage of 50 mm Thick Mongolian Scotch Pine Timber"

_forests, doi:10.3390/f12111500_

Round 1
Reviewer 1 Report
The studies refer to a very specific case, namely the drying of 50 mm thick Mongolian pine. For this study to be very specifically oriented to a single species of a single thickness, this work is too extensive. The relevance of the article is in fact too low, measured by the scope at hand. I recommend shortening the article! In my view, this paper should be shortened, as several results are presented twice. Figure 3 only presents more information than Figure 2, so Figure 2 can be omitted. The same applies to Figure 4, whose meaning I do not understand at all. The tables also contain redundant information. I ask that results be presented only once and not redundantly. The text must be adjusted accordingly. Regardless of this, the styles of the diagrams are very different. There is no uniform graphical representation of all results. It rather looks as if the diagrams were copied together from different works. However, if Figures 2 and 4 are omitted, the remaining diagrams in Figures 3 and 5 can be adapted in style. Otherwise, I assume that the work is original. Unfortunately, the article is very difficult to read. However, the results are described consistently.Author Response
1.The studies refer to a very specific case, namely the drying of 50 mm thick Mongolian pine. For this study to be very specifically oriented to a single species of a single thickness, this work is too extensive
Responses: This is a very important opinion. We studied the shrinkage properties of 50 mm Pinus sylvestris var. mongolica. The methods used and the results obtained can be used as a reference for the wood and other tree species of Mongolian pine.
2.The relevance of the article is in fact too low, measured by the scope at hand;
Responses: Thank you. Your suggestion is very pertinent. This paper first studies the free drying shrinkage coefficient of wood and applies it to practical drying. On this basis, the relevant paragraphs on pages 8 and 13 of the article have been modified to enhance the relevance of the content.
3.I recommend shortening the article! In my view, this paper should be shortened, as several results are presented twice.
Responses: Thank you for pointing out the problems in this article in time. I've shortened the article. Delete the redundant parts in pages 8, 10, etc.
4.Figure 3 only presents more information than Figure 2, so Figure 2 can be omitted.
Responses: Fig. 2 is a free shrinkage curve drawn according to the data obtained during the experiment. Fig. 3 and Fig. 4 are drawn according to the data obtained by curve fitting in Fig. 2. As you pointed out, Fig. 3 and Fig. 4 are highly repetitive. Figure 3 has been deleted and the analysis between them has been simplified.
5.The same applies to Figure 4, whose meaning I do not understand at all.
Responses: For better understanding, I have modified the content here. Figure 4 shows the free drying shrinkage coefficients of different layers in the thickness direction of Mongolian pine at different temperatures. In the experiment, I divided the test material into nine layers from the thickness direction. Due to many data, the similar data have been removed, leaving the relevant data of 1, 5 and 9 representative layers
6.The tables also contain redundant information.
Responses: Table 3 shows the parameters of free drying shrinkage coefficient, total dry shrinkage and fiber saturation point of Mongolian pine test material 1. Table 4 shows the free shrinkage coefficients of the test materials at different heights of the Mongolian pine. As you said, since the relevant data of test material 1 have been shown in Table 3, the relevant data in Table 4 have been deleted.
7.I ask that results be presented only once and not redundantly. The text must be adjusted accordingly.
Responses: Thank you for pointing out the shortcomings of the article. The article order has been modified and redundant content has been deleted. page 8, 10,13,15, etc.
8.Regardless of this, the styles of the diagrams are very different.
Responses: Thank you for your advice. Fig. 3 in the text has been deleted. Part 3.1 in this paper is the experiment of free shrinkage coefficient. The law of free dry shrinkage coefficient between different parts of Mongolian pine in the thickness direction at different temperatures was explored. There are many data, it adopts more intuitive three-dimensional image expression. Part 3.2 is the analysis of the conventional drying free and actual drying shrinkage strain of Mongolian Scotch Pine. Because it is only the ratio between the two in different drying stages, a more appropriate two-dimensional image expression is adopted.
9.There is no uniform graphical representation of all results.
Responses: In order to make the diagrams style close, figure 3 has been removed. The diagrams styles of 3.1 and 3.2 are unified respectively
10.It rather looks as if the diagrams were copied together from different works.
Responses: As mentioned above, the diagrams in this paper correspond to the exploration experiment of free shrinkage coefficient of Mongolian pine in part 3.1 and the analysis of free and actual shrinkage strain of Mongolian Scotch Pine in part 3.2.
11.However, if Figures 2 and 4 are omitted, the remaining diagrams in Figures 3 and 5 can be adapted in style.
Responses: Thank you for your pertinent advice. As mentioned above, I have modified the diagrams style in the paper accordingly.
12.Otherwise, I assume that the work is original.
Responses: The experiment in the paper was completed by myself and has not been published in other journals. Thank you for your timely discovery and pointing out these problems in the process of writing my paper. I have made corresponding modifications to your questions. I hope this paper can be better understood.
13.Unfortunately, the article is very difficult to read. However, the results are described consistently.
Responses: Once again, I apologize for my low level of expression. As mentioned above, this paper is a relatively specific study on the drying strain mechanism of Mongolian pine a. Firstly, the law of free shrinkage coefficient of Mongolian Scotch Pine was explored. Secondly, combined with this law, the stress-strain development process in the conventional drying process is studied (through the comparison of free drying shrinkage and actual drying shrinkage). I have made corresponding modifications in the results and conclusions of the paper.

Reviewer 2 Report
A research paper of great cognitive importance of shrinkage properties of Mongolian Scotch Pine wood. The Authors indicated the degree of shrinkage and stress level in the process of drying changes.
A valuable work, but one that needs to be corrected.
Notes:
Row 124
Table 1 does not contain data describing the FPS parameter and there is an undescribed EMC - equilibrium moisture content
In what directions was the drying shrinkage of Mongolian Scotch Pine tested ? it was tested only in the radial direction- (width of the sample) why was the tangential direction not considered by verifying earlywood and latewood increments?
Table 4 shows the effect of zones along the lumber length on shrinkage ratios. However, the results are for sections extracted from 1 piece of 4 meter lumber. Why did the authors of this study not include a larger length range to verify variable drying shrinkage parameters? - The average height of Mongolian Scotch Pine is approximately 30m. The authors did not specify the harvest zone of the lumber (stub, mid or top of the log?).
The observations in line 245 and following describe the results of tangential and radial shrinkage tests. The authors here did not refer to available publications on Scotch Pine timber.
Line 248
On what basis (no microscopic studies) do the authors jump to changes in cellulose chains (no literature support for presented )?
The paper lacks reference of the results of the conducted research to the literature results (DUSKUSION) which is a shortcoming of the paper.
Conclusions as a summary of the research, could be clarified by giving the values obtained as a result of the experiments.
The Authors could refer to a rich database of publications on the evaluation of changes in wood during drying shrinkage.
(Cave I. D. A theory of the shrinkage of wood. Wood Science and Technology volume 6, pages 284–292 (1972)
Skaar, C. Wood-Water Relations; Springer: Berlin, Germany; New York, NY, USA, 1988; ISBN 0-387-19258-1.
Rowell, R.M. Moisture properties. In Handbook of Wood Chemistry and Wood Composites, 2nd ed.; Rowell, R.M., Ed.; CRC Press: Boca Raton, FL, USA, 2012; pp. 75–98. ISBN 9781439853818.
Etc.)
Author Response
Dear reviewers, I would like to express my sincere thanks for your valuable suggestions. I have answered your questions below:
1.Row 124, Table 1 does not contain data describing the FPS parameter and there is an undescribed EMC - equilibrium moisture content;
Responses: Thanks. I'm sorry at this point. Fiber saturation point (FSP) is my mistake. It is equilibrium moisture content (EMC) and has been modified. I would like to express my highest respect to you for your pertinent comments.
2.In what directions was the drying shrinkage of Mongolian Scotch Pine tested ?
Responses: In this experiment, the chord shrinkage of Pinus sylvestris var. mongolica was measured by chord cutting plate.
3.it was tested only in the radial direction- (width of the sample) why was the tangential direction not considered by verifying earlywood and latewood increments?
Responses: Due to the limitation of traditional slicing method, the experiment mainly measured the chord (width) shrinkage of wood, which has been marked in the article (Page 2.).
4.Table 4 shows the effect of zones along the lumber length on shrinkage ratios. However, the results are for sections extracted from 1 piece of 4 meter lumber. Why did the authors of this study not include a larger length range to verify variable drying shrinkage parameters? - The average height of Mongolian Scotch Pine is approximately 30m. The authors did not specify the harvest zone of the lumber (stub, mid or top of the log?).
Responses: Thank you. This suggestion is very meaningful. The Mongolian Scotch Pine used in this study is limited by transportation and laboratory cold storage, and only the stump is selected; Indicated in the text (Page 4.). Thank you again for pointing out the existing shortcomings, which I will attach great importance to in the future research.
5.The observations in line 245 and following describe the results of tangential and radial shrinkage tests. The authors here did not refer to available publications on Scotch Pine timber.
Responses: For the difference of free shrinkage coefficient in different parts of Mongolian Scotch Pine thickness (near chord and near radial), I have referred to relevant literature and marked it (Page 8.).
6.Line 248, On what basis (no microscopic studies) do the authors jump to changes in cellulose chains (no literature support for presented )?
Responses: Thank you. Here is my reasoning on the difference of wood shrinkage coefficient, which has been modified and marked with references.
7.The paper lacks reference of the results of the conducted research to the literature results (DUSKUSION) which is a shortcoming of the paper.
Responses: Thank you very much. This is a very pertinent opinion. I have supplemented it in the analysis part.
8.Conclusions as a summary of the research, could be clarified by giving the values obtained as a result of the experiments.
Responses: Thanks. I have combined the results and added specific data to the conclusion to support the conclusion.
9.The Authors could refer to a rich database of publications on the evaluation of changes in wood during drying shrinkage.
Responses: Thank you for your recommendation. I have referred to the literature you listed (page 16, reference.2, 22 and 25).

Round 2
Reviewer 1 Report
Thanks for improving the manuscript! I would feel more comfortable if the axis labeling in Fig.2 and 3 could be improved (the letters are too close together), but in my opintion this manuscirpt can be accepted for publication.
Reviewer 2 Report
Work substantially revised.
Errors were removed and additional publications were included.